

# Detection of spreader nodes in human-SARS-CoV protein-protein interaction network

Sovan Saha[1], Piyali Chatterjee[2], Mita Nasipuri[3] and Subhadip Basu[3]

[1] Computer Science and Engineering, Institute of Engineering and Management, Kolkata, West Bengal, India
[2] Computer Science and Engineering, Netaji Subhash Engineering College, Kolkata, West Bengal, India
[3] Computer Science and Engineering, Jadavpur University, Kolkata, West Bengal, India

## ABSTRACT

The entire world is witnessing the coronavirus pandemic (COVID-19), caused by a novel coronavirus (n-CoV) generally distinguished as Severe Acute Respiratory Syndrome Coronavirus 2 (SARS-CoV-2). SARS-CoV-2 promotes fatal chronic respiratory disease followed by multiple organ failure, ultimately putting an end to human life. International Committee on Taxonomy of Viruses (ICTV) has reached a consensus that SARS-CoV-2 is highly genetically similar (up to 89%) to the Severe Acute Respiratory Syndrome Coronavirus (SARS-CoV), which had an outbreak in 2003. With this hypothesis, current work focuses on identifying the spreader nodes in the SARS-CoV-human protein–protein interaction network (PPIN) to find possible lineage with the disease propagation pattern of the current pandemic. Various PPIN characteristics like edge ratio, neighborhood density, and node weight have been explored for defining a new feature *spreadability index* by which spreader proteins and protein–protein interaction (in the form of network edges) are identified. Top spreader nodes with a high *spreadability index* have been validated by Susceptible-Infected-Susceptible (SIS) disease model, first using a synthetic PPIN followed by a SARS-CoV-human PPIN. The ranked edges highlight the path of entire disease propagation from SARS-CoV to human PPIN (up to level-2 neighborhood). The developed network attribute, *spreadability index*, and the generated SIS model, compared with the other network centrality-based methodologies, perform better than the existing state-of-art.

## INTRODUCTION

The COVID-19 pandemic registered its first case on 31 December 2019 (*World Health Organization, 2020b*). First, it laid its foundation in the Chinese city of Wuhan (Hubei province) (*Wang et al., 2020*). Soon, it made several countries worldwide (*Centers for Disease Control and Prevention (CDC), 2021*) its victim by community spreading which ultimately compelled the World Health Organization (*World Health Organization (WHO), 2019*) to declare a global health emergency on 30 January 2020 (*World Health Organization*

Corresponding author
Sovan Saha, sovansaha12@gmail.com

(WHO), 2005b) for the massive outbreak of COVID-19. Owing to its expected fatality rate, which is about 4%, as projected by WHO (*World Health Organization (WHO), 2005a*), researchers from nations all over the world have joined their hands to work together to understand the spreading mechanisms of this virus SARS-CoV-2 (*Heymann, 2020*; *Huang et al., 2020*; *Liu & Wang, 2020*; *Zhou et al., 2020*) and to find out all possible ways to save human lives from the dark shadow of COVID-19.

Coronavirus belongs to the family Coronaviridae. This single-stranded RNA virus affects not only humans but also mammals and birds too. Due to coronavirus, common fever/flu symptoms are noted in humans, followed by acute respiratory infections. Nevertheless, coronaviruses like Middle East Respiratory Syndrome (MERS) and Severe Acute Respiratory Syndrome (SARS) can create a global pandemic due to their infectious nature. Both of these coronaviruses are the member of genus Betacoronavirus under Coronaviridae. SARS started a significant outbreak in 2003, originating from Southern China. Seven hundred seventy-four deaths were reported among 8098 globally registered cases resulting in an estimated fatality rate of 14%–15% (*World Health Organization (WHO), 2003*). While MERS commenced in Saudi Arabia, creating an endemic in 2012. The world witnessed 858 deaths among 2494 registered positive cases. It generated a high fatality rate of 34.4% in comparison to SARS.

SARS-CoV-2 is under the same Betacoronavirus genus as that of MERS and SARS coronavirus (*Lu et al., 2020*). It comprises several structural and non-structural proteins. The structural proteins include the envelope (E) protein, membrane (M) protein, nucleocapsid (N) protein, and the spike (S) protein. Though SARS-CoV-2 has been identified recently, there is an intense scarcity of data and necessary information needed to gain immunity against SARS-CoV-2. Studies have revealed that SARS-CoV-2 is highly genetically similar to SARS-CoV based on several experimental genomic analyses (*Hoffmann et al., 2020*; *Letko, Marzi & Munster, 2020*; *Lu et al., 2020*; *Zhou et al., 2020*). This is also the reason behind the naming of SARS-CoV-2 by the International Committee on Taxonomy of Viruses (ICTV) (*World Health Organization (WHO), 2020a*). Due to this genetic similarity, the immunological study of SARS-CoV may lead to the discovery of SARS-CoV-2 potential drug development.

A protein–protein Interaction Network (PPIN) has been used as the central component in identifying spreader nodes in SARS-CoV in the proposed methodology. PPIN is a very effective module for protein function determination (*Cai, Wang & Deng, 2020*; *Hakala et al., 2020*; *Saha et al., 2019a*; *Saha et al., 2018*; *Saha et al., 2019b*; *Zhao et al., 2020*) as well as in the identification of central/essential spreader nodes in the PPIN (*Anthonisse, 1971*; *He et al., 2021*; *Jeong et al., 2001*; *Joy et al., 2005*; *Li et al., 2011*; *Liu, Ma & Chen, 2019*; *Wen et al., 2020*; *Wuchty & Stadler, 2003*; *Zhong et al., 2021*). The compactness of the PPIN and its transmission capability is estimated using centrality analysis. *Anthonisse (1971)* proposed a new centrality measure named Betweenness Centrality (BC). Another centrality measure, called closeness centrality (CC), is defined by *Sabidussi (1966)*. Two other essential centrality measures: degree centrality (DC) (*Jeong et al., 2001*) and Local average centrality (LAC) (*Li et al., 2011*), are also found to be very effective in this area of research.

Due to the high morbidity and mortality of SARS-CoV2, it has been felt that there is a pressing need to properly understand the way of viral infection transmission from SARS-CoV-2 PPIN to human PPIN. This paper considers SARS-CoV PPIN for this research study due to its high genetic similarity with SARS-CoV-2. Another primary motivation is to study the spreadability pattern of the ancestral strain of nCoV. In the proposed methodology, at first, SARS-CoV-Human PPIN (up to level-2) is formed from the collected datasets (*Agrawal, Zitnik & Leskovec, 2017*; *Pfefferle et al., 2011*). Once created, the spreader nodes are first identified in the SARS-CoV PPIN. Then its level-1 and level-2 interactors in the human PPIN are extracted using a new network attribute, *i.e., spreadability index,* which is a combination of three different network features: (1) edge ratio (*Samadi & Bouyer, 2019*) (2) neighborhood density (*Samadi & Bouyer, 2019*) and (3) node weight (*Wang & Wu, 2013*). The detected spreader nodes in the human PPIN are validated by the Susceptible, Infected, and Susceptible (SIS) epidemic disease model (*Bailey, 1975*). Then the edges connecting two spreader nodes are ranked based on the average *spreadability index.* Thus, the ranked edges highlight the path through which viral infection gets mediated from SARS-CoV to human PPIN (up to level-2). The entire methodology can be categorized into 3-steps for (1) identifying the spreader nodes in the SARS-CoV and human PPIN using *spreadability index*, (2) validation of spreader nodes by SIS model, and (3) ranking of the spreader edges.

Developing the *spreadability index* for raking edges in a host-pathogen PPIN to analyse the host's viral infection propagation path is the primary contribution of this work. Furthermore, considering the current investigation on SARS-CoV and the notable similarity with its successor virus, we also attempt to shed light on the propagation pattern of viral infection of SARS-CoV2 in human PPIN.

In the following, we first describe the theory and methods for different network properties used to extract the PPIN characteristics. Then we describe the 3-step methodology. First, the methodology has been described using a synthetic PPIN (generated by Cytoscape; *Shannon et al., 2003*). Then, in the experimental results section, we have employed the developed method on the human-SARS-CoV PPIN to identify the SARS-CoV viral infection propagation path in the human PPIN. Finally, in the discussion section, we attempt to relate our findings with the ancestral virus, *i.e.,* SARS-CoV, with its successor, *i.e.,* SARS-CoV2, to study the SARS-CoV2 disease propagation may follow the pattern from SARS-CoV.

## THEORY & NOTATIONS

The viral infection gets mediated from one part of the PPIN to another through spreader nodes and edges (*Brito & Pinney, 2017*). Generally, in disease-specific PPIN models, at least two entities are involved: pathogen/Bait and host/Prey (*Saha et al., 2017*). In this research work, SARS-CoV takes the role of the former while human the latter one. Viral proteins of SARS-CoV tend to target their corresponding interaction with human proteins, which target its next level of proteins. So, the establishment of interactions between SARS-CoV and human occurs through connected nodes and edges of PPIN. But mostly, these viral
proteins try to interact more with the central/hub proteins rather than the other proteins (*Brito & Pinney, 2017*). Thus, proper identification of central nodes (*i.e.,* spreader nodes) is required. It is also confirmed that the interaction is not possible without the edges connecting two spreader nodes. Thus, these connecting edges are called spreader edges. The proposed methodology involves a proper study and assessment of various existing established PPIN features followed by identifying spreader nodes, which the SIS model has also verified. Before going into the detailed study about the proposed work, various network-based terminologies which are used in this work are discussed below:

## 1. Protein–protein interaction network (PPIN)

When one protein interacts with another protein, it forms a network-like structure known as PPIN. Generally, it is portrayed as a graph where proteins are represented as nodes, and their corresponding connecting edges represent their interactions. Mathematically, PPIN can be highlighted as a graph $G_{nv}$, which consists of a set of vertices $v$(nodes) connected by edges $e$ (links). Thus, $G_{nv} = (v, e)$ (*Saha et al., 2014*; *Saha et al., 2019a*).

## 2. Level-1 and Level-2 proteins

In a PPIN, level-1 proteins of a node are those proteins that are in direct connection with that node, *i.e.,* its immediate neighbors, whereas level-2 proteins are those proteins that are indirectly connected with level-1 proteins of that node, *i.e.,* its indirect neighbors (*Saha et al., 2014*; *Saha et al., 2019a*).

## 3. Graph centrality

Graph centrality is one of the essential aspects for the identification of significant nodes in a PPIN. The centrality of a node defines how relevant the node is in a PPIN or how much a node is centrally located in a PPIN.

## 4. Betweenness centrality (BC)

BC (*Anthonisse, 1971*) is one of the ways of measuring a node's impact on the transmission of information between every pair of nodes in a graph, considering that this transmission is always executed over the shortest path between them. Mathematically, it is defined as:

$$C_B^{(u)} = \sum_{s \neq u \neq t} \frac{\rho(s, u, t)}{\rho(s, t)}$$

where $\rho(s, t)$ is the total number of shortest paths from node $s$ to node $t$, and $\rho(s, u, t)$ is the number of those paths that pass through $u$.

## 5. Closeness centrality (CC)

CC (*Sabidussi, 1966*) is a procedure for detecting nodes that transmit information within a network efficiently. Nodes with high closeness centrality values are considered to have the shortest distance to all available nodes in the network. It can be mathematically expressed as:

$$C_C^{(u)} = \frac{|N_u| - 1}{\sum_{v \in V} dist(u, v)}$$

where $|N_u|$ denotes the number of neighbors of node $u$ and $dist(u, v)$ is the distance of the shortest path from node $u$ to node $v$.

## 6. Degree centrality (DC)

DC (*Jeong et al., 2001*) is considered the simplest among the available centrality measures that only count the degree of a node, *i.e.,* the number of directly connected neighbors. Nodes having a high degree are said to be the highly connected module of the network. It is defined as:

$$C_D^{(u)} = |N_u|$$

where $|N_u|$ denotes the number of neighbors of node $u$.

## 7. Local average centrality (LAC)

LAC (*Li et al., 2011*) of a node represents how close its neighborhood proteins are. It is defined to be the local metric to compute the essentiality of the node for transmission ability by considering its modular nature, the mathematical model of which is highlighted as:

$$LAC(u) = \frac{\sum_{w \in N_u} deg_{C_u}^w}{|N_u|}$$

where $C_u$ is the subgraph induced by $N_u$ (*i.e.,* the number of neighbors of node $u$) and $deg_{C_u}^w$ is the total number of nodes that are directly connected in $C_u$.

## 8. Ego network

Ego network of node $i$ ($S_i$) (*Samadi & Bouyer, 2019*) is defined as the grouping of node $i$ itself along with its corresponding level-1 neighbors and interconnections. N ($S_i$) (*Samadi & Bouyer, 2019*) consists of the set of nodes which belong to the ego network, $S_i$ *i.e.,* $\{i\} \cup \Gamma(i)$.

## 9. Edge ratio

The edge ratio of node $i$ (*Samadi & Bouyer, 2019*) is defined by the following equation:

$$Edge\ ratio\ (i) = \frac{\left(\sum_{j \in (i)} |\Gamma(j) - N(S_i)|\right) + 1}{\left(\frac{1}{2} \sum_{j \in \Gamma(j)} |\Gamma^{S_i}(J)|\right) + 1} = \frac{E_{out}^{S_i} + 1}{E_{in}^{S_i} + 1}$$

where $E_{out}^{S_i}$ is the total number of interactions between the ego network $S_i$ and the proteins outside it. $E_{in}^{S_i}$ is the total number of interactions among node $i$'s neighbors. $\Gamma(i)$ denotes the level-1 neighbors of node $i$. $S_i$ is considered to be Ego network. $\Gamma^{S_i}(j)$ denotes node $j$'s neighbors which belongs $S_i$. In the edge ratio, $E_{out}^{S_i}$ is positively related to the non-peripheral location of node $i$. A large number of interactions resulting from the ego network denotes that the node has a high level of interconnectivity between its neighbors. On the other hand, $E_{in}^{S_i}$ is negatively related to the inter-module location of node $i$. It represents the fact that the interconnectivity between neighbors is usually connected to the number of structural holes available around the node. Thus, when the neighbor's interconnectivity is low, the root or the central node $i$ gains more control of transmission flow among the neighbors.

## 10. Jaccard dissimilarity

The similarity between two nodes is determined by Jaccard dissimilarity (*Jaccard, 1912*) based on their common neighbors. Jaccard dissimilarity of node $i$ and $j$ ($dissimilarity(i,j)$) is defined as:

$$dissimilarity\ (i,j) = 1 - sim(i,j) = \frac{|\Gamma(i) \cap \Gamma(j)|}{|\Gamma(i) \cup \Gamma(j)|}$$

where $|\Gamma(i) \cap \Gamma(j)|$ refers to the number of common neighbors of $i$ and $j$. $|\Gamma(i) \cup \Gamma(j)|$ is the total number of neighbors of $i$ and $j$. The similarity degree between $i$ and $j$ is considered more when they have more common neighbors. Whereas, when dissimilarity between the neighbors of a node is high, it guarantees that the only common node among the neighbors is the central node, which is termed a structural hole situation (*Samadi & Bouyer, 2019*).

## 11. Neighborhood diversity

The neighborhood diversity (*Samadi & Bouyer, 2019*) is a significant parameter of a graph that is based on Jaccard dissimilarity. When the dissimilarity of the neighbors of a node is high, it assures that the central node is the only neighbor common among the neighbors of that node, *i.e.,* it represents the structural hole situation. On the other hand, when a node's neighborhood diversity reaches its greatest value, it reveals that the neighbors have no other closer path. Hence, the neighbors should transmit or communicate through this node. Mathematically, it is defined as:

$$neighborhood\_diversity\ (i) = \sum_{j,k \in (i)} dissimilarity\ (j,k).$$

## 12. Node weight

Node weight (*Wang & Wu, 2013*) is a graph parameter used to assign weightage to a node in a graph. Node weight $w_v$ of node $v \in V$ in PPIN is interpreted as the average degree of all nodes in $G_{V'}$, a sub-graph of a graph $G_V$. It is considered as another measure to determine the strength of connectivity of a node in a network. Mathematically, it is represented by

$$w_v = \frac{\sum_{u \in V''} \deg(u)}{|V''|}$$

where $V''$ is the set of nodes in $G_{V'}$. $|V''|$ is the number of nodes in $G_{V'}$. And $deg(u)$ is the degree of a node $u \in V''$.

## DATASET

Three datasets are mainly used for the present study. They are (1) SARS-CoV PPIN (*Pfefferle et al., 2011*) which contains only interactions of viral SARS-CoV proteins. (2) SARS-CoV-Human PPIN (*Pfefferle et al., 2011*) contains interaction information of SARS-CoV and human proteins. (3) Human PPIN (*Agrawal, Zitnik & Leskovec, 2017*; *BioSNAP, 2021*), which contains only interactions of human proteins. These datasets are mainly used to generate two types of PPIN: (1) Synthetic PPIN and (2) Biological PPIN. Synthetic PPINs

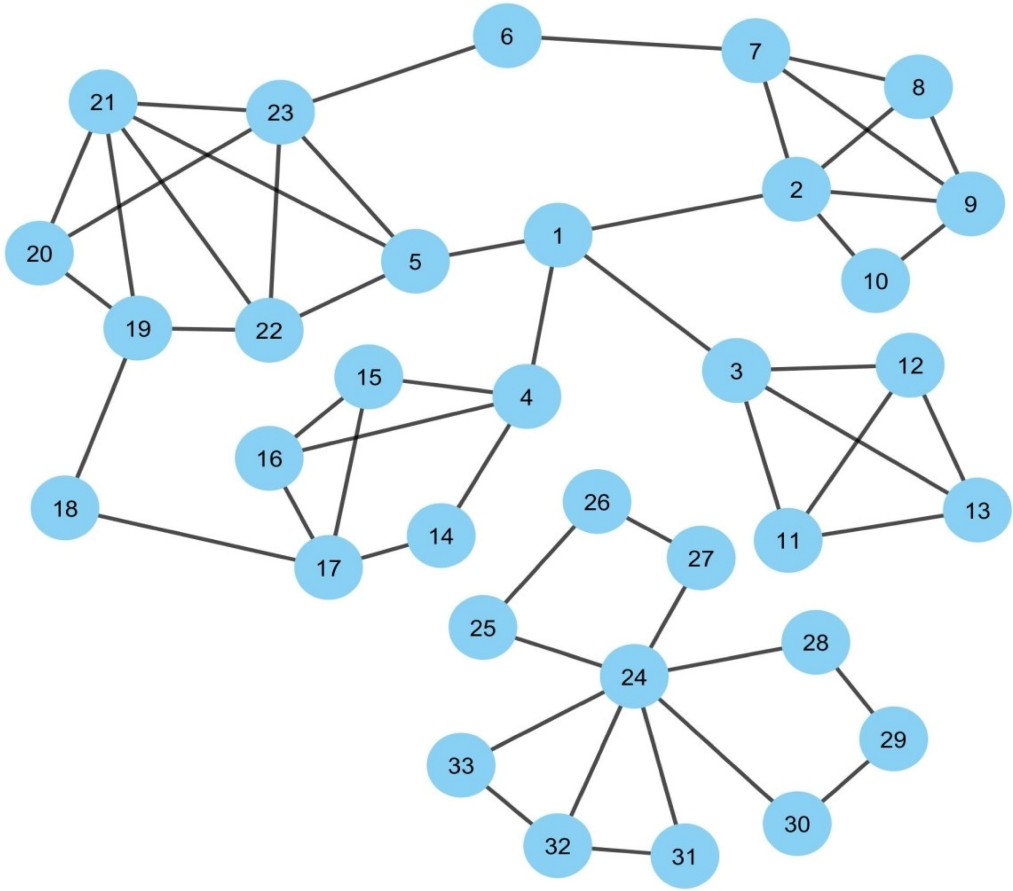

**Figure 1 Synthetic PPIN1.** The PPIN consists of 33 nodes and 53 edges. Nodes 1, 24 are the essential spreaders. Node 1 connects the four densely connected modules of the PPIN, which turns this node to stand in the first position having the highest spreadability index. Node 24 holds the second position for the spreadability index. Node 24 is one of the most densely connected modules itself despite getting isolated from the main PPIN module of node 1.

are the randomly generated sample PPINs (nodes with edges) used for the detailed analysis and testing of the proposed methodology (for example, please see Fig. 1). The algorithm of the same is discussed in the supplementary document. Biological PPINs are the complete PPINs generated from the above datasets on which the proposed methodology is executed after testing (for example, please the complete PPIN view of SARS-CoV and human PPIN added at the end of the Experimental Results and Discussion section).

## METHODOLOGY

The proposed work can be mainly categorized into three sub-sections: (1) Identification of spreader nodes by *spreadability index*, (2) Validation of spreader nodes by SIS model, and (3) Ranking of spreader edges.

## 1. Identification of spreader nodes by spreadability index

The *spreadability index* of node *i* is defined as the ability of node *i* to mediate a viral infection in a PPIN. Mathematically it can be defined as:

$$Spreadability\_index(i) = \big(Edgeratio\ (i) \times neighborhood\_diversity\ (i)\big) + Nodeweight\ (w_i).$$

Nodes having a high *spreadability index* are termed as spreader nodes, *i.e.,* if the viral proteins establish interactions with these nodes, then the viral infection can be mediated to a more significant number of nodes in a much short amount of time compared to the other nodes in PPIN.

Figure 1 represents a sample PPIN where each protein is denoted as a node while edges mark its interactions with other proteins. The PPIN consists of 33 nodes and 53 edges. The PPIN data and the protein names and interactions are given as input to the Cytoscape, which generates the network view as highlighted in Fig. 1. Cytoscape is open-source software that is used for PPIN generation and visualization (*Shannon et al., 2003*). The spreadability index is computed on the synthetic PPIN, shown in Fig. 1, using essential PPIN characteristics in this PPIN, as stated earlier. The same is compared to DC, BC, CC, and LAC, highlighted in Tables 1 to 5.

In Fig. 1, it can be observed that nodes 1 and 24 are the essential spreaders. Node 1 connects the four densely connected modules of the PPIN, making this node the topper with the highest spreadability index. This node has been correctly ranked by all the methods except LAC and DC. Node 24, though, has a moderate edge ratio and node weight but is one of the most densely connected modules itself despite getting isolated from the main PPIN module of node 1. Moreover, node 24 has the highest neighborhood density. It establishes that the only path of transmission of information for nodes 26, 27, 25, 28, 29, 30, 31, 32, and 33 is node 24. Thus, if viral proteins of SARS-CoV establishes interaction with node 24, then all the connected nodes will be indirectly coming under the interaction of viral proteins as the connected nodes have no interactions with other central nodes except node 24. So, node 24 holds the second position for the *spreadability index* in our proposed methodology. Node 24 is not correctly identified as the second most influential spreader node by the other methods. Further assessment of the remaining nodes highlights the fact that the performance of the new attribute *spreadability index* in our proposed methodology is relatively better in comparison to the others.

## 2. Validation of spreader nodes by SIS model

To design the mathematical model for this infectious disease, the SIS Epidemic Model (*Bailey, 1975*) is used in this proposed methodology by classifying the proteins in SARS-CoV-human PPIN based on their interactivity status (for more details, please see "*Studied Models in epidemiology*" section of the supplementary document). SIS refers to **S**usceptible, **I**nfected and **S**usceptible states, which are generally considered the three probable protein states in a PPIN. (1) **S** - The susceptible states are the states of those human proteins with which viral proteins have not yet interacted, but they are at risk of getting interacted. In general, every protein in PPIN is initially in a susceptible state. (2) **I** – These infected states are the states of those human proteins with which viral proteins have interacted, and the

**Table 1 Computation of spreadability index of synthetic Fig. 1 and computation of spreadability rate of selected top 10 spreader nodes by the SIS model.**

| Rank | Proteins | $E^{S_i}_{out}$ | $E^{S_i}$ | Edge ratio | Neighborhood diversity | Node weight | Spreadability index | Sum of SIS spreadability rate of top 10 nodes |
|---|---|---|---|---|---|---|---|---|
| 1 | 1 | 13 | 0 | 14.0 | 5.19 | 3.40 | 76.15 | |
| 2 | 24 | 4 | 2 | 1.66 | 12.5 | 1.87 | 22.70 | |
| 3 | 4 | 6 | 1 | 3.50 | 3.63 | 2.40 | 15.11 | |
| 4 | 5 | 8 | 3 | 2.25 | 4.39 | 3.60 | 13.48 | |
| 5 | 19 | 6 | 2 | 2.33 | 3.8 | 2.80 | 11.66 | 2.46 |
| 6 | 23 | 5 | 4 | 1.20 | 6.58 | 3.00 | 10.89 | |
| 7 | 17 | 4 | 1 | 2.50 | 3.33 | 2.00 | 10.33 | |
| 8 | 6 | 7 | 0 | 8.00 | 0.87 | 3.00 | 10.00 | |
| 9 | 2 | 4 | 4 | 1.00 | 6.84 | 2.83 | 9.68 | |
| 10 | 22 | 6 | 4 | 1.40 | 3.88 | 3.60 | 9.03 | |
| 11 | 25 | 7 | 0 | 8.00 | 0.71 | 3.00 | 8.71 | |
| 12 | 27 | 7 | 0 | 8.00 | 0.71 | 3.00 | 8.71 | |
| 13 | 28 | 7 | 0 | 8.00 | 0.71 | 3.00 | 8.71 | |
| 14 | 30 | 7 | 0 | 8.00 | 0.71 | 3.00 | 8.71 | |
| 15 | 18 | 6 | 0 | 7.00 | 0.85 | 2.66 | 8.66 | |
| 16 | 20 | 7 | 2 | 2.66 | 1.78 | 3.50 | 8.26 | |
| 17 | 7 | 4 | 3 | 1.25 | 4.15 | 2.80 | 7.98 | |
| 18 | 21 | 3 | 6 | 0.57 | 6.66 | 3.33 | 7.13 | |
| 19 | 3 | 3 | 3 | 1.00 | 4.00 | 2.60 | 6.60 | |
| 20 | 16 | 4 | 2 | 1.66 | 2.06 | 2.75 | 6.19 | |
| 21 | 15 | 4 | 2 | 1.66 | 2.06 | 2.75 | 6.19 | _ |
| 22 | 31 | 6 | 1 | 3.50 | 0.75 | 3.33 | 5.95 | |
| 23 | 33 | 6 | 1 | 3.50 | 0.75 | 3.33 | 5.95 | |
| 24 | 32 | 4 | 2 | 1.66 | 1.75 | 2.75 | 5.66 | |
| 25 | 8 | 4 | 3 | 1.25 | 1.88 | 3.25 | 5.60 | |
| 26 | 14 | 6 | 0 | 7.00 | 0.40 | 2.66 | 5.46 | |
| 27 | 9 | 2 | 4 | 0.60 | 3.64 | 2.80 | 4.98 | |
| 28 | 10 | 5 | 1 | 3.00 | 0.50 | 3.00 | 4.50 | |
| 29 | 13 | 1 | 3 | 0.50 | 1.70 | 2.50 | 3.35 | |
| 30 | 11 | 1 | 3 | 0.50 | 1.70 | 2.50 | 3.35 | |
| 31 | 12 | 1 | 3 | 0.50 | 1.70 | 2.50 | 3.35 | |
| 32 | 29 | 2 | 0 | 3.00 | 0.00 | 1.33 | 1.33 | |
| 33 | 26 | 2 | 0 | 3.00 | 0.00 | 1.33 | 1.33 | |

viral infection gets mediated. (3) **S** –The susceptible states are the states of those human proteins that have lost their interaction with the viral proteins (due to antiviral therapies or change in interface residues (*Brito & Pinney, 2017*)) and again become susceptible. The interaction rate of the viral proteins with human proteins, the loss rate of interactivity of the human protein with the viral proteins (general assumption is that any protein after coming out of the infected state gets into a susceptible state again in one day), and the

**Table 2  Computation of CC of synthetic Fig. 1 and computation of spreadability rate of selected top 10 spreader nodes by the SIS model.**

| Rank | Proteins | Closeness centrality | Sum of SIS spreadability rate of top 10 nodes |
|------|----------|----------------------|-----------------------------------------------|
| 1 | 1 | 0.085 | |
| 2 | 5 | 0.083 | |
| 3 | 2 | 0.082 | |
| 4 | 4 | 0.082 | |
| 5 | 23 | 0.081 | 1.94 |
| 6 | 3 | 0.081 | |
| 7 | 21 | 0.081 | |
| 8 | 22 | 0.081 | |
| 9 | 7 | 0.08 | |
| 10 | 15 | 0.08 | |
| 11 | 16 | 0.08 | |
| 12 | 19 | 0.079 | |
| 13 | 14 | 0.079801 | |
| 14 | 9 | 0.079602 | |
| 15 | 20 | 0.079602 | |
| 16 | 6 | 0.079602 | |
| 17 | 8 | 0.079404 | |
| 18 | 17 | 0.078818 | |
| 19 | 10 | 0.078624 | |
| 20 | 11 | 0.078049 | |
| 21 | 12 | 0.078049 | – |
| 22 | 13 | 0.078049 | |
| 23 | 18 | 0.07767 | |
| 24 | 24 | 0.041558 | |
| 25 | 32 | 0.041237 | |
| 26 | 28 | 0.041237 | |
| 27 | 30 | 0.041237 | |
| 28 | 25 | 0.041237 | |
| 29 | 27 | 0.041237 | |
| 30 | 31 | 0.041184 | |
| 31 | 33 | 0.041184 | |
| 32 | 29 | 0.040921 | |
| 33 | 26 | 0.040921 | |

total number of proteins are usually provided as input to SIS model. If a protein gets into an infected state and has many neighbors, any neighbor can mediate viral infection. So, the final result is generated after 50 iterations for each protein in the infected state. The total number of proteins in the susceptible state after 50 iterations in the neighborhood of each protein in an infected state divided by the total number of proteins in the PPIN gives the interaction capability of the protein in an infected state. Thus, the spreader nodes

**Table 3** Computation of BC of synthetic Fig. 1 and computation of spreadability rate of selected top 10 spreader nodes by the SIS model.

| Rank | Proteins | Betweeness centrality | Sum of SIS spreadability rate of top 10 nodes |
|------|----------|------------------------|-----------------------------------------------|
| 1 | 1 | 269.1 | |
| 2 | 2 | 117.93 | |
| 3 | 4 | 117.1 | |
| 4 | 3 | 114 | |
| 5 | 5 | 108 | 2.2 |
| 6 | 24 | 57 | |
| 7 | 23 | 56.4 | |
| 8 | 19 | 45.56 | |
| 9 | 17 | 39.1 | |
| 10 | 7 | 36.9 | |
| 11 | 6 | 32.9 | |
| 12 | 18 | 32 | |
| 13 | 21 | 29.36 | |
| 14 | 22 | 20.53 | |
| 15 | 16 | 12.1 | |
| 16 | 15 | 12.1 | |
| 17 | 14 | 12.1 | |
| 18 | 28 | 7 | |
| 19 | 30 | 7 | |
| 20 | 25 | 7 | |
| 21 | 27 | 7 | – |
| 22 | 20 | 6.63 | |
| 23 | 9 | 4.16 | |
| 24 | 32 | 1 | |
| 25 | 29 | 1 | |
| 26 | 26 | 1 | |
| 27 | 8 | 0 | |
| 28 | 11 | 0 | |
| 29 | 12 | 0 | |
| 30 | 13 | 0 | |
| 31 | 10 | 0 | |
| 32 | 31 | 0 | |
| 33 | 33 | 0 | |

identified by the *spreadability index* are validated by the interaction rate as generated by the SIS model for them. It can be observed from Tables 1 to 5 that the proposed methodology has the highest SIS interaction rate of 2.46 with viral proteins (see Table 1) in comparison to others for their corresponding top 10 spreader nodes in the synthetic PPIN, as shown in Fig. 1.

**Table 4  Computation of LAC of synthetic Fig. 1 and computation of spreadability rate of selected top 10 spreader nodes by the SIS model.**

| Rank | Proteins | Local average centrality | Sum of SIS spreadability rate of top 10 nodes |
|------|----------|--------------------------|-----------------------------------------------|
| 1 | 21 | 2.4 | |
| 2 | 9 | 2 | |
| 3 | 22 | 2 | |
| 4 | 8 | 2 | |
| 5 | 11 | 2 | 2.19 |
| 6 | 12 | 2 | |
| 7 | 13 | 2 | |
| 8 | 2 | 1.6 | |
| 9 | 23 | 1.6 | |
| 10 | 7 | 1.5 | |
| 11 | 3 | 1.5 | |
| 12 | 5 | 1.5 | |
| 13 | 16 | 1.33 | |
| 14 | 15 | 1.33 | |
| 15 | 20 | 1.33 | |
| 16 | 32 | 1.33 | |
| 17 | 19 | 1 | |
| 18 | 10 | 1 | |
| 19 | 31 | 1 | |
| 20 | 33 | 1 | |
| 21 | 24 | 0.57 | – |
| 22 | 4 | 0.5 | |
| 23 | 17 | 0.5 | |
| 24 | 1 | 0 | |
| 25 | 14 | 0 | |
| 26 | 18 | 0 | |
| 27 | 6 | 0 | |
| 28 | 28 | 0 | |
| 29 | 29 | 0 | |
| 30 | 30 | 0 | |
| 31 | 25 | 0 | |
| 32 | 26 | 0 | |
| 33 | 27 | 0 | |

## 3. Ranking of Spreader edges

To show the ranking of interacting spreader edges, two synthetic PPINs: PPIN-1 and PPIN-2, have been considered in Fig. 2. Node D, E, and F are the selected top spreader nodes in PPIN-1 by *spreadability index,* similarly explained with a synthetic PPIN in Fig. 1. To avoid the complexity in the diagram, the top 5 nodes in PPIN-2 (see Table 1) are selected as spreader nodes. Red-colored edges are the interconnectivity within PPIN-1,

**Table 5** Computation of DC of synthetic **Fig. 1** and computation of spreadability rate of selected top 10 spreader nodes by the SIS model.

| Rank | Proteins | Degree centrality | Sum of SIS spreadability rate of top 10 nodes |
|---|---|---|---|
| 1 | 24 | 7 | |
| 2 | 2 | 5 | |
| 3 | 23 | 5 | |
| 4 | 21 | 5 | |
| 5 | 1 | 4 | 2.3 |
| 6 | 7 | 4 | |
| 7 | 9 | 4 | |
| 8 | 3 | 4 | |
| 9 | 4 | 4 | |
| 10 | 17 | 4 | |
| 11 | 5 | 4 | |
| 12 | 22 | 4 | |
| 13 | 19 | 4 | |
| 14 | 8 | 3 | |
| 15 | 11 | 3 | |
| 16 | 12 | 3 | |
| 17 | 13 | 3 | |
| 18 | 16 | 3 | |
| 19 | 15 | 3 | |
| 20 | 20 | 3 | |
| 21 | 32 | 3 | – |
| 22 | 10 | 2 | |
| 23 | 14 | 2 | |
| 24 | 18 | 2 | |
| 25 | 6 | 2 | |
| 26 | 28 | 2 | |
| 27 | 29 | 2 | |
| 28 | 30 | 2 | |
| 29 | 25 | 2 | |
| 30 | 26 | 2 | |
| 31 | 27 | 2 | |
| 32 | 31 | 2 | |
| 33 | 33 | 2 | |

while black-colored edges show the interconnectivity within PPIN-2. Green-colored spreader edges (*i.e.,* edges connected with spreader nodes) show the interconnectivity between PPIN-1 and PPIN-2. Ranking of a spreader edge measures the interaction ability of a spreader edge with the viral proteins, *i.e.,* how many nodes get interacted with the viral proteins through that edge, and the viral infection gets mediated. Thus, all the spreading

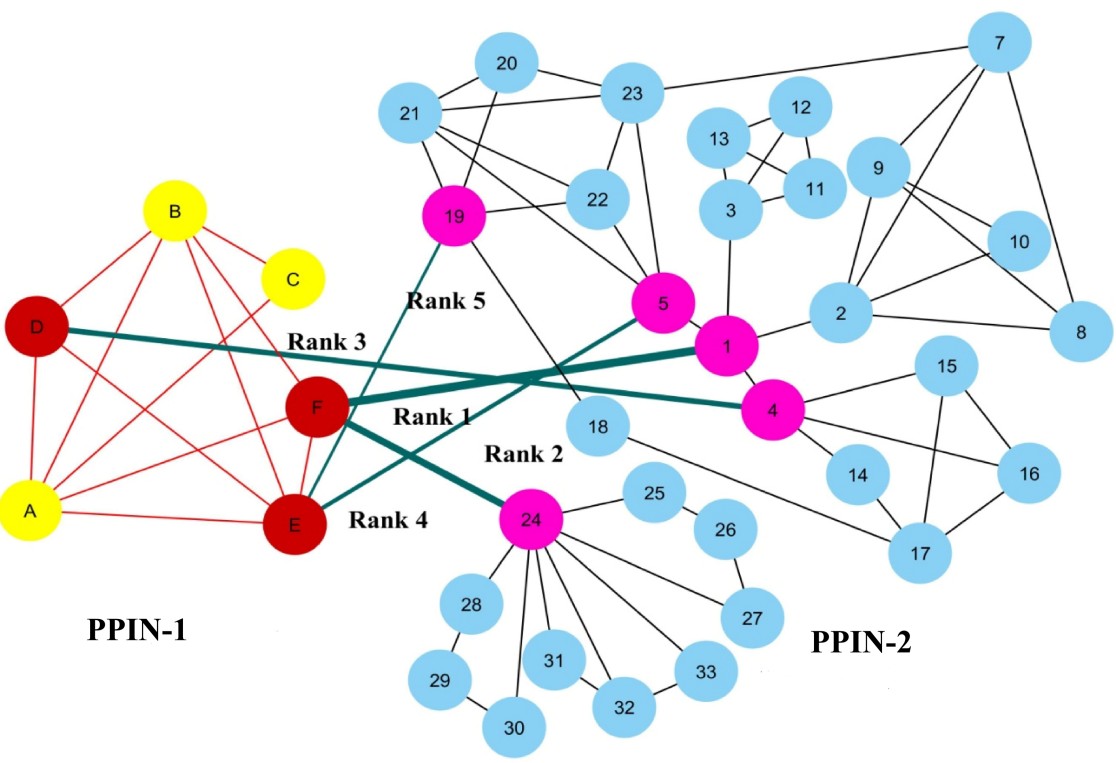

**Figure 2  Ranking of spreader edges.** Two synthetic PPINs: PPIN-1 and PPIN-2, have been considered for ranking spreader edges based on the spreadability index. Red-colored edges are the interconnectivity within PPIN-1, while black-colored edges show the interconnectivity within PPIN-2. Nodes D, E and F, are the detected spreader nodes of PPIN-1, whereas nodes 1, 4, 5, 19 and 24 are the detected spreader nodes of PPIN2. Green-colored spreader edges (*i.e.,* edges connected with spreader nodes) show the interconnectivity between PPIN-1 and PPIN-2. The thickness of the edges varies with the order of ranking.

**Table 6  Ranking of spreader edges for PPIN-1 and PPIN-2 in Fig. 2.**

| Rank | Spreader edges | | Spreadability index of spreader nodes in network 1 | Spreadability index of spreader nodes in network 2 | Ranking of spreader edges |
|---|---|---|---|---|---|
| | Spreader nodes in network 1 | Spreader nodes in network 2 | | | |
| 1 | F | 1 | 5.5 | 76.15 | 40.825 |
| 2 | F | 24 | 5.5 | 22.70 | 14.104 |
| 3 | D | 4 | 5.5 | 15.11 | 10.308 |
| 4 | E | 5 | 4.7 | 13.48 | 9.0919 |
| 5 | E | 19 | 4.7 | 11.66 | 8.1833 |

edges are ranked based on the average *spreadability index* of its connected spreader nodes. The ranked spreader edges in Fig. 2 are highlighted in Table 6.

## EXPERIMENTAL RESULTS & DISCUSSION

The proposed methodology leads to the identification of spreader nodes and edges through a network characteristic, called spreader index which has also been checked and validated

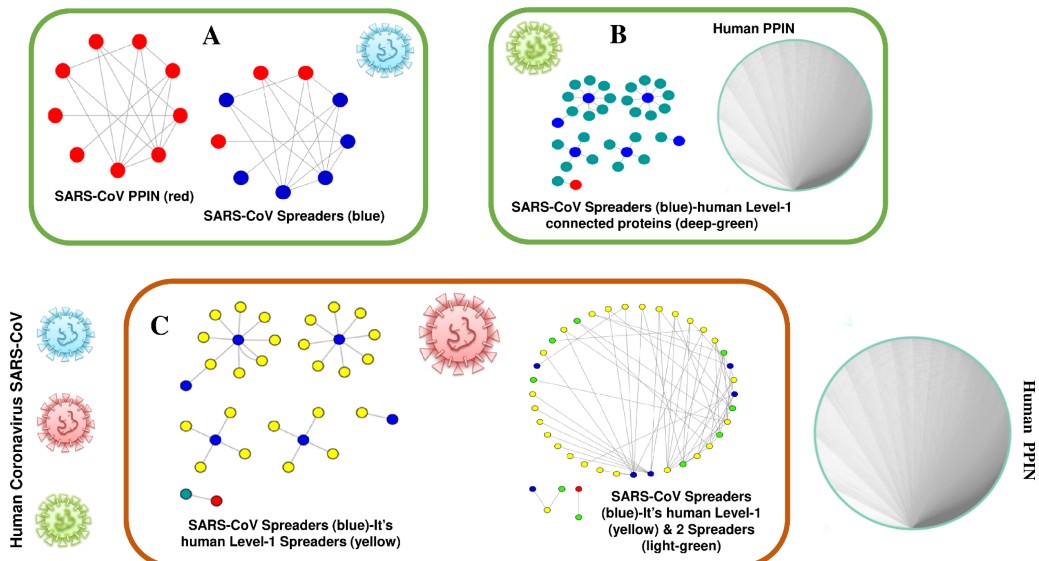

**Figure 3 Mechanism of transmission of viral infection.** SARS-CoV spreaders establish interaction with human spreader proteins, and the viral infection gets mediated from SARS-CoV PPIN to human PPIN through them. (A) PPIN of SARS-CoV (red) in which spreader nodes are marked as blue. (B) Interaction of SARS-CoV spreaders with its level-1 corresponding proteins in human PPIN (marked as green). (C) Selection of spreaders in level-1 (level-1 spreaders are marked as yellow) and level-2 human proteins (level-2 spreaders are marked as green). Rest proteins in human PPIN are ignored to prevent overlap in the diagram.

by the SIS model. Initially, the whole working module is implemented on synthetic PPINs, as shown in the Methodology section, and then on the SARS-CoV-human dataset. For this proposed methodology, three PPIN datasets have been curated, already stated in the dataset section. After removing self-loops and data redundancy, the final SARS-CoV PPIN consists of 17 interactions among 7 SARS-CoV unique proteins (proteins having only one frequency of occurrence). Only the densely interconnected SARS-CoV proteins having direct connections (level-1) with human proteins are considered rather than isolated proteins. SARS-CoV-Human PPIN includes 118 interactions between SARS-CoV and humans. It is used to fetch the level-1 interaction of human proteins for the corresponding SARS-CoV proteins in SARS-CoV PPIN. Human PPIN consists of 314,384 interactions. It is utilized for getting the indirect interactions (level-2) of level-1 human proteins formed earlier. The application of the proposed methodology in SARS-CoV-human PPIN is highlighted in Fig. 3. In Fig. 3A, at first, SARS-CoV PPIN is displayed in which each protein is marked in red. After that, spreader nodes in SARS-CoV PPIN are identified by the *spreadability index.* They are denoted as blue nodes among the red. Once the spreader nodes are active (Fig. 3B), the viral infection gets mediated through its corresponding direct partners, *i.e.,* human-level-1proteins (marked in deep green). Then, in Fig. 3C, spreader nodes are identified in SARS-CoV level-1 human proteins (marked in yellow). The same will continue to SARS-CoV level-2 human proteins (light green nodes are the spreaders).
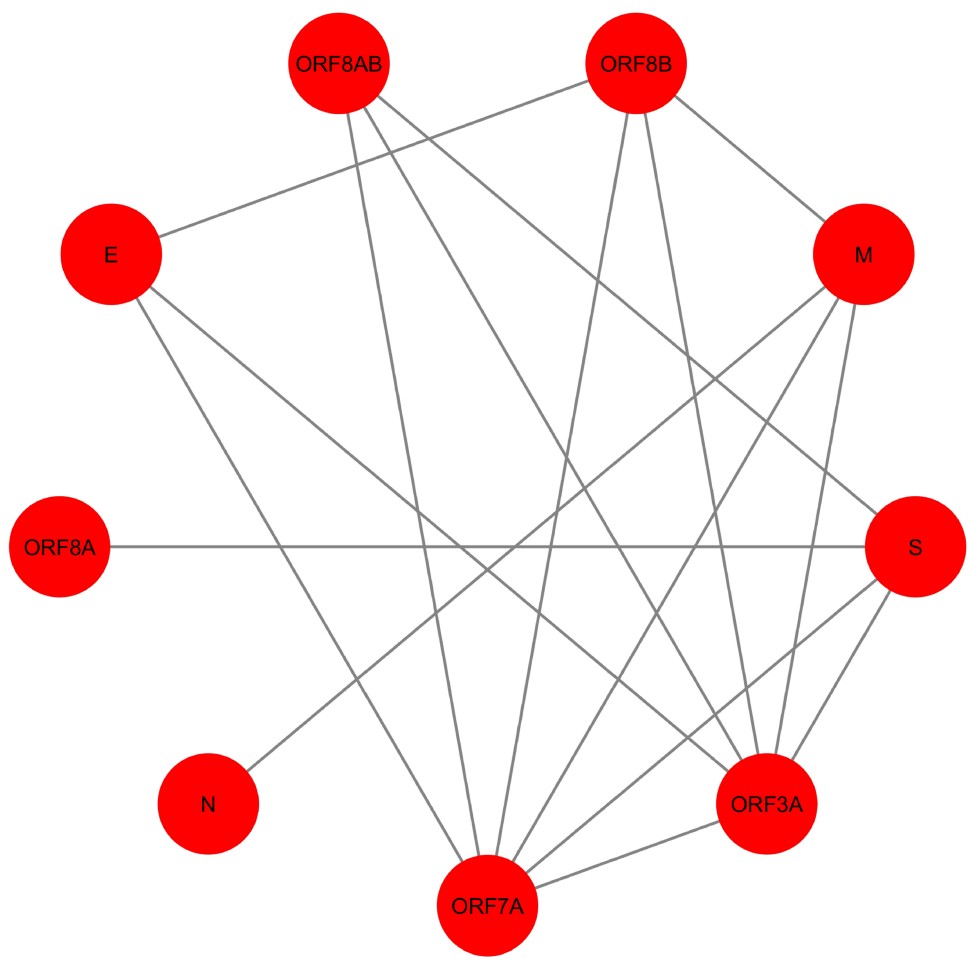

**Figure 4** **SARS-CoV PPIN.** The PPIN consists of the interaction between SARS-CoV proteins only. It is a collection of 9 SARS-CoV proteins only (marked as red).

**Table 7** Computation of spreadability index of SARS-CoV PPIN and computation of spreadability rate of selected top 6 spreader nodes by the SIS model.

| Rank | Proteins | $E_{out}^{S_i}$ | $E^{S_i}$ | Edge ratio | Neighborhood density | Node weight | Spreadability index | SIS spreadability rate of top 6 nodes | Sum of SIS spreadability rate of top 6 nodes |
|------|----------|------|------|------|------|------|------|------|------|
| 1 | M | 7 | 3 | 2.0 | 3.845 | 3.4 | 11.090 | 1 | |
| 2 | S | 6 | 3 | 1.75 | 4.047 | 3.2 | 10.283 | 0.2 | |
| 3 | ORF8AB | 7 | 3 | 2.0 | 1.785 | 4.0 | 7.5714 | 1 | |
| 4 | ORF8B | 5 | 5 | 1.0 | 3.464 | 3.8 | 7.2642 | 0.2 | 2.935 |
| 5 | E | 7 | 3 | 2.0 | 1.428 | 4.0 | 6.8571 | 0.25 | |
| 6 | ORF3A | 2 | 8 | 0.333 | 9.249 | 3.428 | 6.5119 | 0.285 | |
| 7 | ORF7A | 2 | 8 | 0.333 | 9.25 | 3.428 | 6.5119 | – | – |
| 8 | ORF8A | 3 | 0 | 4.0 | 0.0 | 2.0 | 2 | | |
| 9 | N | 3 | 0 | 4.0 | 0.0 | 2.0 | 2 | | |

**Table 8** Computation of degree centrality of SARS-CoV PPIN and computation of spreadability rate of selected top six spreader nodes by the SIS model.

| Rank | Proteins | Degree centrality | SIS spreadability rate of top 6 nodes | Sum of SIS spreadability rate of top 6 nodes |
|------|----------|-------------------|---------------------------------------|----------------------------------------------|
| 1 | ORF7A | 6 | 0.285 | |
| 2 | ORF3A | 6 | 0.285 | |
| 3 | ORF8B | 4 | 0.2 | |
| 4 | M | 4 | 0.6 | 1.82 |
| 5 | S | 4 | 0.2 | |
| 6 | E | 3 | 0.25 | |
| 7 | ORF8AB | 3 | – | – |
| 8 | N | 1 | | |
| 9 | ORF8A | 1 | | |

**Table 9** Computation of closeness centrality of SARS-CoV PPIN and computation of spreadability rate of selected top six spreader nodes by the SIS model.

| Rank | Proteins | Closeness centrality | SIS spreadability rate of top 6 nodes | Sum of SIS spreadability rate of top 6 nodes |
|------|----------|----------------------|---------------------------------------|----------------------------------------------|
| 1 | ORF7A | 0.239 | 0.285 | |
| 2 | ORF3A | 0.239 | 0.285 | |
| 3 | ORF8B | 0.224 | 0.2 | |
| 4 | M | 0.224 | 0.6 | 1.82 |
| 5 | S | 0.224 | 0.2 | |
| 6 | E | 0.215 | 0.25 | |
| 7 | ORF8AB | 0.22 | – | – |
| 8 | N | 0.196 | | |
| 9 | ORF8A | 0.196 | | |

In Fig. 4, SARS-CoV PPIN has been highlighted. There are mainly nine proteins, including E, M, ORF3A, ORF7A, S, N, ORF8A, ORF8AB, and ORF8B. The computed *spreadability index* of these proteins and the corresponding validation by the SIS model are highlighted in Table 7. It is also compared with other central/ influential spreader node detection methodologies like DC, CC, LAC, and BC, shown in Tables 8–11. Similarly, spreader nodes are also identified in SARS-CoV's level-1 neighbors and level-2 neighbors (see Figs. 5 and 6).

The *spreadability index* plays a vital role in this proposed methodology. Spreader nodes are successfully identified by this scoring technique which covers all the aspects through which viral infection gets mediated from one node to another in a PPIN (*Brito & Pinney, 2017*). It should be mentioned here that while identifying spreader nodes in SARS-CoV level-2 human proteins, it has been noted that the number of nodes is getting increased significantly with the increment of successive levels. So, high, medium, and low thresholds

**Table 10  Computation of local average centrality of SARS-CoV PPIN and computation of spreadability rate of selected top six spreader nodes by the SIS model.**

| Rank | Proteins | Local average centrality | SIS spreadability rate of top 6 nodes | Sum of SIS spreadability rate of top 6 nodes |
|------|----------|-------------------------|---------------------------------------|----------------------------------------------|
| 1 | ORF7A | 2.666 | 0.285 | |
| 2 | ORF3A | 2.666 | 0.285 | |
| 3 | ORF8B | 2.5 | 0.2 | |
| 4 | E | 2 | 0.25 | 2.22 |
| 5 | ORF8AB | 2 | 1 | |
| 6 | S | 1.5 | 0.2 | |
| 7 | M | 1.5 | – | – |
| 8 | N | 0 | | |
| 9 | ORF8A | 0 | | |

**Table 11  Computation of betweeness centrality of SARS-CoV PPIN and computation of spreadability rate of selected top six spreader nodes by the SIS model.**

| Rank | Proteins | Betweeness centrality | SIS spreadability rate of top 6 nodes | Sum of SIS spreadability rate of top 6 nodes |
|------|----------|----------------------|---------------------------------------|----------------------------------------------|
| 1 | M | 14 | 0.6 | |
| 2 | S | 14 | 0.2 | |
| 3 | ORF7A | 13.33 | 0.285 | |
| 4 | ORF3A | 13.33 | 0.285 | 1.82 |
| 5 | ORF8B | 1.33 | 0.2 | |
| 6 | E | 0 | 0.25 | |
| 7 | ORF8AB | 0 | – | – |
| 8 | N | 0 | | |
| 9 | ORF8A | 0 | | |

(*Zhang et al., 2016*) have been applied, and the entire viral infection mediation through *spreadability index* is computationally assessed at each threshold. The network statistics of spreader nodes at each level of threshold are shown in Table 12. It can be observed that threshold application is only implemented at SARS-CoV level-2 human proteins, not on others. This is because of the availability of a smaller number of nodes and edges. Therefore, only nodes and edges having a shallow *spreadability index* have been discarded at the first level.

Besides the identification of spreader nodes, spreader edges are also identified. The ranked edges between SARS-CoV spreaders and its level-1 human spreaders are highlighted in Table 13. In contrast, the ranked edges between SARS-CoV s level-1 and level-2 human spreaders at high, medium, and low thresholds are highlighted in the Tables S1–S3, respectively. The supplementary document is available online here. The complete PPIN

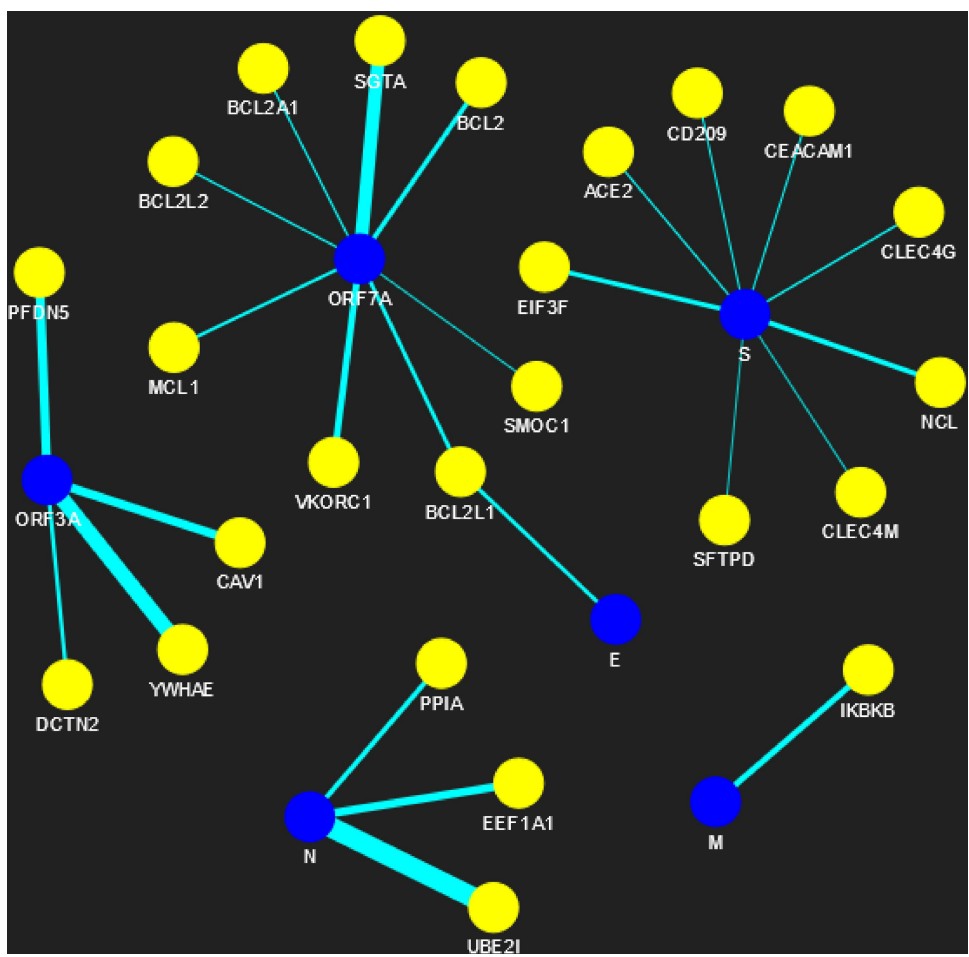

**Figure 5** **SARS-CoV-human PPIN (level-1).** The PPIN consists of the interaction between SARS-CoV and human proteins. The blue node represents SARS-CoV spreaders, while the yellow node represents SARS-CoV s level-1 human spreaders. The thickness of the edges varies with the order of ranking.

view of SARS-CoV and human PPIN has been generated online (by using the pyvis module available in python) under three circumstances:

(1) All the nodes and edges are considered spreader nodes and edges respectively and ranked accordingly.

https://yu2qkp7gwoinjwsebyw0xw-on.drv.tw/www.graph_all.html/graph_all.html.

(2) Selected Spreader nodes and edges are highlighted for the high threshold.

https://yu2qkp7gwoinjwsebyw0xw-on.drv.tw/www.high_threshold.com/graph_high_threshold.html.

(3) Selected Spreader nodes and edges are highlighted for the medium threshold.

https://yu2qkp7gwoinjwsebyw0xw-on.drv.tw/www.medium_threshold.com/graph_medium_threshold.html.

(4) Selected Spreader nodes and edges are highlighted for the low threshold.

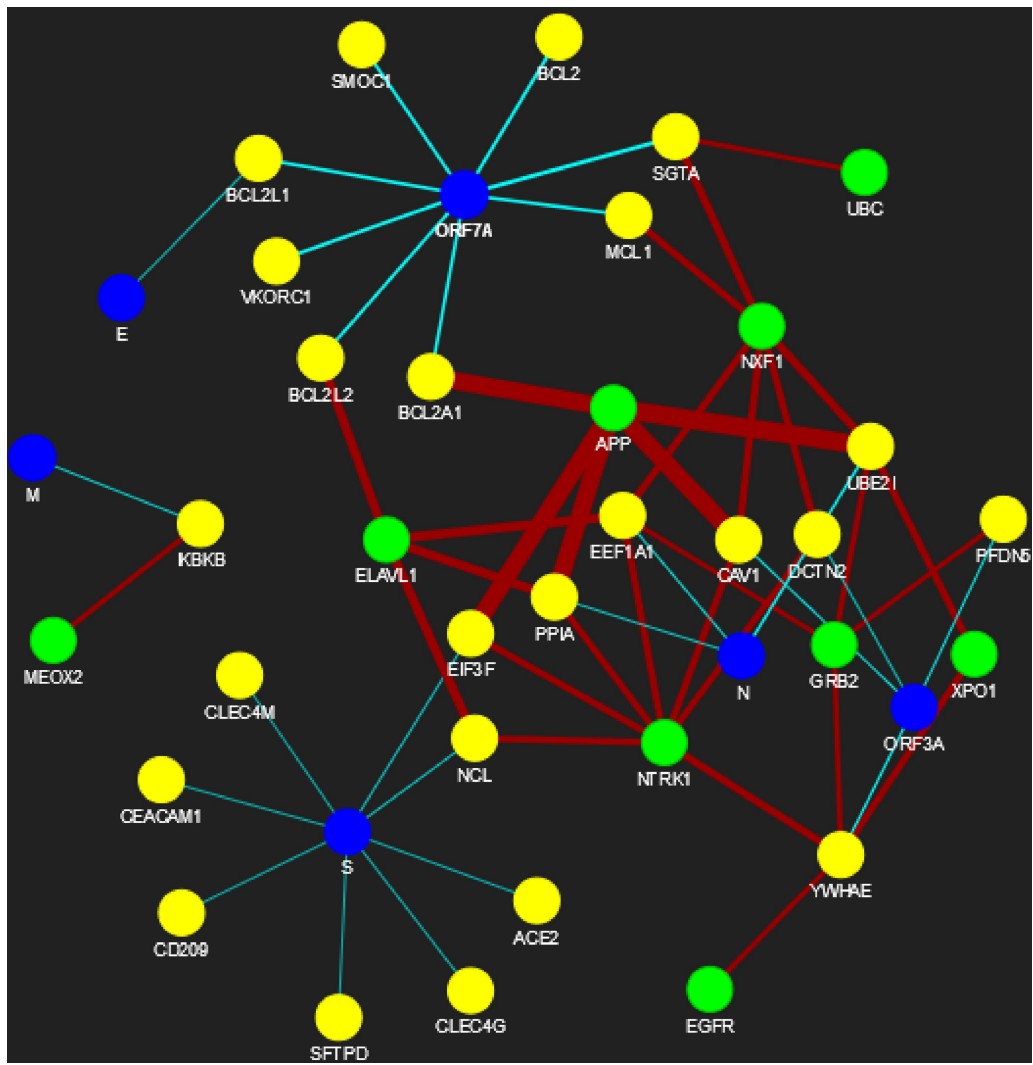

**Figure 6** **SARS-CoV-human PPIN (level-1 and level-2).** The PPIN consists of the interaction between SARS-CoV and human proteins. The blue node represents SARS-CoV spreaders, while the yellow and green nodes represent SARS-CoV s level-1 and level-2 human spreaders. The thickness of the edges varies with the order of ranking.

https://yu2qkp7gwoinjwsebyw0xw-on.drv.tw/www.low_threshold.com/graph_low_threshold.html.

In the above-generated PPIN views, the blue, yellow, and green colors represent SARS-CoV spreaders, level-1 human spreaders, and its level2 human spreaders. The remaining nodes are in indigo.

## CONCLUSION

The *spreadability index* is thus proved to be effective in detecting spreader nodes and edges in SARS-CoV-human PPIN and the cross-validation by the SIS model. Spreader nodes are the central nodes in the PPIN through which viral infection gets mediated to

**Table 12  Network statistics of spreaders at three levels of thresholds.**

| Threshold | SARS-CoV spreaders | SARS-CoV-s level 1 human spreaders | SARS-CoV-s level 2 human spreaders |
|---|---|---|---|
| High | 6 | 24 | 9 |
| Medium | 6 | 24 | 22 |
| Low | 6 | 24 | 111 |

**Table 13  Ranked spreader edges between SARS-CoV spreaders and its level-1 human spreaders.**

| | Spreader edges | | |
|---|---|---|---|
| Rank | SARS-CoV spreaders | SARS-CoV s level 1 human spreaders | Spreading ability of spreader edges |
| 1 | N | UBE2I | 679697.677 |
| 2 | ORF3A | YWHAE | 500684.2755 |
| 3 | ORF7A | SGTA | 428397.3206 |
| 4 | ORF3A | PFDN5 | 273863.194 |
| 5 | ORF3A | CAV1 | 264566.0653 |
| 6 | N | EEF1A1 | 241407.2776 |
| 7 | ORF7A | VKORC1 | 187916.2768 |
| 8 | M | IKBKB | 164728.3002 |
| 9 | S | NCL | 131643.7345 |
| 10 | N | PPIA | 125719.6427 |
| 11 | S | EIF3F | 119529.0273 |
| 12 | ORF7A | BCL2 | 119299.092 |
| 13 | ORF3A | DCTN2 | 92293.0019 |
| 14 | E | BCL2L1 | 89404.47117 |
| 15 | ORF7A | BCL2L1 | 89404.29855 |
| 16 | ORF7A | MCL1 | 63953.80825 |
| 17 | S | CLEC4G | 27477.4133 |
| 18 | ORF7A | BCL2L2 | 22974.97399 |
| 19 | ORF7A | BCL2A1 | 22252.28441 |
| 20 | S | ACE2 | 18775.88601 |
| 21 | S | CEACAM1 | 14834.82402 |
| 22 | S | CD209 | 12215.99362 |
| 23 | ORF7A | SMOC1 | 6068.990602 |
| 24 | S | CLEC4M | 3844.528751 |
| 25 | S | SFTPD | 119.09278 |

their successors. Simultaneously, if the spreader nodes are not connected with spreader edges, that would not have been possible. In a nutshell, it can be said that the proposed work exploits the possibility of understanding how viral infection gets mediated from the SARS-CoV PPIN to the human PPIN. It should be borne in mind that SARS-CoV2 is ~89% genetically similar to its predecessor SARS-CoV (*Chan et al., 2020*; *CIDRAP, 2020*). Therefore, it strongly reveals that the human proteins chosen as spreaders of SARS-CoV might be the potential targets of SARS-CoV2. So, the same concept of the

*Spreadability index* is applied along with a unique fuzzy protein–protein interaction model to form SARS-CoV2-human PPIN in our other research work (*Saha et al., 2020a*). The formed PPIN is also compared (*Saha et al., 2020b*) with that of SARS-CoV2-Human PPIN generated in the work of Gordon et al. (*Gordon et al., 2020*). Henceforth, study and analysis of drug repurposing of COVID-19 are also implemented in the subsequent research work (*Saha et al., 2020b*). Thus, it explores a new direction in identifying essential drugs/vaccines for SARS-CoV2. Recently, the work is limited to only SARS-CoV/SARS-CoV2, which can be further extended to other viral infectious diseases in our future work.

### Funding

The authors received support (infrastructure facilities) from the "Center for Microprocessor Applications for Training Education and Research" research laboratory of the Computer Science and Engineering Department, Jadavpur University, India. In addition, this project is also supported by the Department of Biotechnology project (No. BT/PR16356/BID/7/596/2016), Ministry of Science and Technology, Government of India. There was no additional external funding received for this study. The funders had no role in study design, data collection and analysis, decision to publish, or preparation of the manuscript.

### Grant Disclosures

The following grant information was disclosed by the authors:
Center for Microprocessor Applications for Training Education and Research.
Department of Biotechnology Project: BT/PR16356/BID/7/596/2016).
Ministry of Science and Technology, Government of India.

### Competing Interests

The authors declare there are no competing interests.

### Author Contributions

- Sovan Saha and Subhadip Basu conceived and designed the experiments, performed the experiments, analyzed the data, prepared figures and/or tables, authored or reviewed drafts of the paper, and approved the final draft.
- Piyali Chatterjee and Mita Nasipuri conceived and designed the experiments, analyzed the data, prepared figures and/or tables, authored or reviewed drafts of the paper, and approved the final draft.

### Data Availability

The data and code are available at GitHub: https://github.com/SovanSaha/Detection-of-spreader-nodes-in-Human-SARS-CoV-protein-protein-interaction-network.

## Supplemental Information

Supplemental information for this article can be found online at http://dx.doi.org/10.7717/peerj.12117#supplemental-information.

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
