# Peer review of "Detection of spreader nodes in human-SARS-CoV protein-protein interaction network"

_PeerJ, doi:10.7717/peerj.12117_

## Round 0.1 · original submission · Major Revisions

I agree with the comments and suggestions provided by the reviewers. In particular, more appropriate terms or at least describing most network-related terms should be considered. Also, a brief description of the synthetic network construction should be provided.

Reviewer 1 ·

Basic reporting

The manuscript is well written and easy to read. Background material is presented in a clear and concise way. Figures and tables are of high quality.

Experimental design

The research question is well defined. Methods are described in full detail.

Validity of the findings

The findings are valid, and can be easily replicated. However, see the general comments below.

Additional comments

The authors focus their study on the SARS-CoV protein-protein interaction network because of the high genetic similarity of SARS-CoV and SARS-CoV-2, but the protein-protein interaction network of SARS-CoV-2 has also been studied (10.1038/s41586-020-2286-9). The manuscript will be much more meaningful by performing the same analysis on the SARS-CoV-2 protein-protein interaction network as well. This would allow to draw further conclusions about the pattern of disease propagation from virus to human.

·

Basic reporting

This is an interesting and important study that adds significantly to the field and will have a noticeable impact.

Although the manuscript is written in a professional English, in my view an excessive usage of "transmission of infection", "transmit the infection", "infected proteins", "susceptible proteins which are not yet infected but are at risk of getting infected", "proteins that have recovered and again become susceptible: and other alike terms is misleading. Protein cannot be infected. A node in PPIN cannot transmit infection. Therefore, statement on lines 282-284 "the infection will penetrate further in human PPIN resulting in a significant fall in human immunity level followed by the severe acute respiratory syndrome" is misleading and confusing. Although it is clear what the authors want to say here, the used terminology is wrong, as in reality they are considering transmission of information, and not infection. Therefore, all related parts of the manuscript should be rewritten to make ideas less ambiguous.

Also, it is highly advisable to provide definitions of all the network-related terms (e.g., centrality, degree, node weight, neighborhood density, etc.) in a layman language. In fact, readers not familiar with graph theory would be completely lost.

Experimental design

Experimental design is appropriate.

Validity of the findings

Findings are interesting and important, but their description uses wrong terminology and therefore is very confusing.

Additional comments

The idea of introduction of the spreadability index for the analysis of the transmission of information within SARS-CoV and human PPINs and from SARS-CoV PPIN to human PPIN is very interesting and promising. Data generated as the result of the implementing of this measure are useful. However, the terminology used for the description of the work is misleading and confusing.

Reviewer 3 ·

Basic reporting

The article provides sufficient background information. The theory that was used was clearly stated. But there are several points in the article that need attention.
- Several outdated references, e.g. in line 81-88. The authors might need to add the updated references.
- Line 108-111: redundant information with line 91- 102.
- Line 176: typo error, “hole”.

Experimental design

The authors need to improve in the methods section.
- The author should add the data collection part. Even though it is noted in the results section, it is necessary to clarify how the data for this study was gathered.
- How did the synthetic network was constructed? What are the data (protein and interaction) that have been used to construct the synthetic network?
- Line 248-250: this sentence is confusing. The author might either use PPIN or network, instead of using both PPIN and network. Term of network or PPIN should be consistently used throughout the manuscript.

Validity of the findings

- The results section need to be improved and organized, can organize the results into subsections.
- Several terms should be adequately defined, such as level-1, level-2, unique proteins.
- All figure captions should clearly describe the figures. For example, the authors should add the explanation on the spreader node in Figure 1. The figure caption for figure 1 and figure 4 sound alike.
- Figure 3 should be rearranged. Figure A and B should be placed on top of Figure C. The caption is confusing. The meaning of the node colors is unclear. Typo error on the caption and the label of the figure.
- Figure 4 is the SARS-CoV network or SARS-CoV-human network? The difference between SARS-CoV and SARS-CoV-human PPIN should be described in the manuscript.
- Line 285: redundant terms. PPIN of the SARS-CoV network. It might be either SARS-CoV PPIN or SARS-CoV network.
- Line 308-309: the complete network has been generated online. Specific tool and parameters used to construct the network should be mentioned in the methods section.
- Discussion part should be elaborated more. For example, how do the most spreader nodes (at least top 3 or top 5 spreader nodes) relate to COVID19. How do the network in Figure 4, 5 and 6 relate to COVID19.
- Can you highlight the significance of the network, proteins and edges identified in this study with COVID19?
- How does the developed network attribute, spreadability index, and the generated SIS model perform better than existing approaches? How do you know SIS model performs better than other approaches? @as any testing carried out? Please explain.
- Discussion and conclusion parts can be separated.

Additional comments

The background is well structured and written. However, the authors need to improve on other sections, as followed:
- Please improve on figures and all figure captions.
- There are grammatical mistakes and typo errors throughout the manuscript.
- Several points, such as data collection and the network construction (tools, parameters in constructing the network), need to be addressed in the methods.
- The organization of results need to be improved.
- The results need to be critically discussed and concluded.
- Please be consistent with the term of PPIN and network. Several terms need to be clearly defined.

---

## Round 0.2 · accepted · Accept

I agree with the reviewers that the authors addressed all the raised questions and the current version of the manuscript is acceptable for publication in PeerJ.

·

Basic reporting

No comments

Experimental design

No comments

Validity of the findings

No comments

Additional comments

All the critiques are adequately addressed and the manuscript is revised accordingly. I am please to see that terminology was fixed and definitions of all the network-related terms were added. I am satisfied with revision and do not have new suggestions.

Reviewer 3 ·

Basic reporting

The article provides sufficient background information. The theory that was used was clearly stated.

Experimental design

The experimental design was well defined. The authors have improved a lot in describing the experimental design.

Validity of the findings

The findings and conclusions are well stated.

Additional comments

Authors have made a full effort to revise the manuscript. I suggest accepting this manuscript.

---

## Author Rebuttal · Round 0.2

# Editor/Reviewer's Response

## Editor's Comment:

*1. I agree with the comments and suggestions provided by the reviewers. In particular, more appropriate terms or at least describing most network-related terms should be considered. Also, a brief description of the synthetic network construction should be provided.*

**Reply:** Full effort has been made to revise the entire manuscript according to the reviewer's comments. It is now updated with more relevant and related terminologies. Description of the synthetic PPIN construction is now included as Algorithm1 in the updated supplementary. Please see Section 1 of the supplementary document. The synthetic PPIN view is developed using a PPIN construction tool named Cytoscape, which has been highlighted in the methodology section of the manuscript. All changes are marked in red.

## Reviewer-1 Response:

*1. The authors focus their study on the SARS-CoV protein-protein interaction network because of the high genetic similarity of SARS-CoV and SARS-CoV-2, but the protein-protein interaction network of SARS-CoV-2 has also been studied (10.1038/s41586-020-2286-9). The manuscript will be much more meaningful by performing the same analysis on the SARS-CoV-2 protein-protein interaction network as well. This would allow us to draw further conclusions about the pattern of disease propagation from virus to human.*

**Reply:** Thank you for your valuable suggestion. The current work focuses only on identifying spreader nodes and edges in SARS-CoV PPIN because of the high genetic similarity of SARS-CoV and SARS-CoV-2. But with the gradual progress, the same has also been applied to form SARS-CoV2-Human PPIN in our next research work, which is available as a preprint in arXiv [1]. Here, human spreader nodes are identified by the spreadability index in the SARS-CoV-human interaction PPIN, which is also validated by Susceptible-Infected-Susceptible (SIS) disease model. This serves as an input to a unique fuzzy protein-protein interaction model along with UniProt collected n-CoV proteins. It generates level 1 human spreaders of n-CoV. Hence its level 2 human spreaders are identified using the spreadability index as implemented earlier. Once the SARS-CoV2-Human PPIN is formed, drug repurposing of COVID-19 has also been extensively studied in another work of ours which is also available as a preprint in OSF [2]. The same PPIN has also been compared [2] with the SARS-CoV2-Human PPIN generated in the work of Gordon et al. (10.1038/s41586-020-2286-9) [3]. All the above information is now added in the Conclusion section. Please see the lines marked in red.

### References:

1. S. Saha, A.K. Halder, S.S. Bandyopadhyay, P. Chatterjee, M. Nasipuri, S. Basu, Computational modeling of Human nCoV protein-protein interaction network, ArXiv (2020). https://arxiv.org/abs/2005.04108
2. Saha, S., Halder, A. K., Bandyopadhyay, S. S., Chatterjee, P., Nasipuri, M., Bose, D., & Basu, S. (2020, May 12). Is Fostamatinib a possible drug for COVID-19? – A computational study. https://doi.org/10.31219/osf.io/7hgpj
3. Gordon DE, Jang GM, Bouhaddou M, Xu J, Obernier K, White KM, et al. A SARS-CoV-2 protein interaction map reveals targets for drug repurposing. Nature 2020; 583:459–68. https://doi.org/10.1038/s41586-020-2286-9.

## Reviewer-2 Response:

*1. This is an interesting and important study that adds significantly to the field and will have a noticeable impact. Although the manuscript is written in professional English, in my view an excessive usage of "transmission of infection", "transmit the infection", "infected proteins",*

*"susceptible proteins which are not yet infected but are at risk of getting infected", "proteins that have recovered and again become susceptible: and other alike terms is misleading. Protein cannot be infected. A node in PPIN cannot transmit infection. Therefore, the statement on lines 282-284 "the infection will penetrate further in human PPIN resulting in a significant fall in human immunity level followed by the severe acute respiratory syndrome" is misleading and confusing. Although it is clear what the authors want to say here, the used terminology is wrong, as in reality, they are considering the transmission of information and not infection. Therefore, all related parts of the manuscript should be rewritten to make the ideas less ambiguous. Also, it is highly advisable to provide definitions of all the network-related terms (e.g., centrality, degree, node weight, neighborhood density, etc.) in a layman language. In fact, readers not familiar with graph theory would be completely lost.*

**Reply:** Thank you for highlighting this issue. According to Brito et al. [1], "*Viral infections are mediated by several protein–protein interactions (PPIs), which can be represented as networks (protein interaction networks, PINs), with proteins being depicted as nodes, and their interactions as edges. It has been suggested that viral proteins tend to establish interactions with more central and highly connected host proteins. In an evolutionary arms race, viral and host proteins are constantly changing their interface residues, either to evade or to optimize their binding capabilities. Apart from gaining and losing interactions via rewiring mechanisms, virus–host PINs also evolve via gene duplication (paralogy); conservation (orthology); horizontal gene transfer (HGT) (xenology); and molecular mimicry (convergence)*". Considering this, a full effort has been made to correct the wrong terminologies not to create further confusion. Related sections in the manuscript have been updated. Please see the lines marked in red. In addition, basic terminologies like PPIN, level-1, level-2, graph centrality, etc., have been newly added to make them better understandable. In contrast, the existing ones are updated into simpler and more generalized forms. For details, please see the Theory & Notations section. Updated sections have been highlighted in red.

**References**

1. Brito AF, and Pinney JW. 2017. Protein–Protein Interactions in Virus–Host Systems. *Frontiers in Microbiology* 8:1557.

*2. Findings are interesting and important, but their description uses wrong terminology and therefore is very confusing. The idea of introduction of the spreadability index for the analysis of the transmission of information within SARS-CoV and human PPINs and from SARS-CoV PPIN to human PPIN is very interesting and promising. Data generated as the result of the implementing of this measure are useful. However, the terminology used for the description of the work is misleading and confusing.*

**Reply:** Thank you for your words of appreciation. A complete effort has been made to correct the wrong terminologies, not to create further confusion. Related sections in the manuscript have been updated. Please see the lines marked in red.

**Reviewer-3 Response:**

*1. The article provides sufficient background information. The theory that was used was clearly stated. But there are several points in the article that need attention. Several outdated references, e.g. in line 81-88. The authors might need to add the updated references.*
*Line 108-111: redundant information with line 91- 102.*
*Line 176: typo error, "hole".*

**Reply:** Thank you for highlighting this issue. The references are updated in lines 81-88. Redundant information between Line 108-111 and line 91- 102 is now removed. Please see the lines marked in red. According to the work of Samadi et al. [1], "*structural hole situation*" is a particular scenario in a graph that results when dissimilarity between the neighbors of a node is high. This high dissimilarity guarantees that the only common node among the neighbors is the central node.

**References:**

1. Samadi, N., Bouyer, A. Identifying influential spreaders based on edge ratio and neighborhood diversity measures in complex networks. Computing 101, 1147–1175 (2019). https://doi.org/10.1007/s00607-018-0659-9

*2. The authors need to improve in the methods section.*
*- The author should add the data collection part. Even though it is noted in the results section, it is necessary to clarify how the data for this study was gathered.*
*- How did the synthetic network was constructed? What are the data (protein and interaction) that have been used to construct the synthetic network?*
*- Line 248-250: this sentence is confusing. The author might either use PPIN or network, instead of using both PPIN and network. Term of network or PPIN should be consistently used throughout the manuscript.*

**Reply:** Thank you for your valuable, insightful suggestion. Dataset section has been added before methodology section. Synthetic PPINs are the randomly generated sample PPINs (nodes with edges) used for the detailed analysis and testing of the proposed method. The same is also compared to DC, BC, CC, and LAC, as shown in Table 1 to Table 5. Description of the synthetic PPIN construction is now included as Algorithm1 in the updated supplementary. Please see Section 1 of the supplementary document. The algorithm is also added here below for your ready reference. The section "Ranking of Spreader edges" (Line 248-250) and Figure 2 are updated with the term PPIN only instead of the network. All the changes have been marked as red in the manuscript.

| **Algorithm 1:** Synthetic PPIN Formation |
|---|
| **Input**: PPIN represented by an undirected graph where each vertex represents a protein and edge represents the interactions.<br>No. of nodes (k) |
| **Output**: Synthetic PPIN with given no. of nodes (k) |

Begin
*//formation of a list of unique nodes*
  M = ∅//Let M be an empty list
  k = 0
// i ≠ j and 1<=i, j<=n, n is the total no. of proteins in PPIN
  for each combination of protein pair $(P_i, P_j)$ in PPIN
      split the protein pair $(P_i, P_j)$
      append each protein $P_i$ and $P_j$ to M
      k = k + 1
   end for
*//Formation of unique set of proteins by applying set ()*
   set M=list(set(M))
 *// Random selection of nodes by importing random module and storing in random_nodes*
   set random_nodes to random.sample(nodes,k)
 *//Display selected random nodes*

```
    output (random_nodes)
//Display of interactions of selected random nodes i.e., synthetic PPIN
    for each protein $P_i$ in random_nodes
        for each combination of protein pair $(P_i,P_j)$ in PPIN
            if protein $P_i$ present in protein pair $(P_i,P_j)$
                output (protein pair $(P_i,P_j)$)
            end if
        end for
    end for
End
```

***3. The results section needs to be improved and organized, can organize the results into subsections. Several terms should be adequately defined, such as level-1, level-2, unique proteins.***

**Reply:** Experimental Results section is now thoroughly updated. PPIN, centrality, Level-1, and level-2 terminologies have been now adequately defined in the Theory & Notations section for better understandability. In addition, unique proteins explanation is added in the Experimental Results section. Updated areas have been highlighted in red.

***4. All figure captions should clearly describe the figures. For example, the authors should add the explanation on the spreader node in Figure 1. The figure caption for figure 1 and figure 4 sound alike.***
***- Figure 3 should be rearranged. Figure A and B should be placed on top of Figure C. The caption is confusing. The meaning of the node colors is unclear. Typo error on the caption and the label of the figure.***
***- Figure 4 is the SARS-CoV network or SARS-CoV-human network? The difference between SARS-CoV and SARS-CoV-human PPIN should be described in the manuscript.***
***- Line 285: redundant terms. PPIN of the SARS-CoV network. It might be either SARS-CoV PPIN or SARS-CoV network.***

**Reply:** All the figure labels are updated. In Figure 3, Figures A and B are now placed above Figure C. The caption of Figure 3 is changed along with the labels inside the Figure for better representation of node colors. The corresponding text in the manuscript is also updated. Changes are marked in red. The revised figures and captions are also given below for your ready reference. SARS-CoV PPIN consists of only interactions of SARS-CoV proteins, whereas SARS-CoV-human PPIN consists of interactions between SARS-CoV and human proteins. The same details are also included in the newly added Dataset section (please see the lines marked in red). Redundant terms in Line 285 are now removed.

[Figure]

Figure 1 **Synthetic PPIN.** The PPIN consists of 33 nodes and 53 edges. Nodes 1, 24 are the essential spreaders. Node 1 connects the four densely connected modules of the PPIN, which turns this node to stand in the first position having the highest spreadability index. Node 24 holds the second position for the spreadability index. Node 24 is one of the most densely connected modules itself despite getting isolated from the main PPIN module of node 1.

[Figure]

Figure 2 **Ranking of Spreader edges.** Two synthetic PPINs: PPIN-1 and PPIN-2, have been considered for ranking spreader edges based on the spreadability index. Red-colored edges are the interconnectivity within PPIN-1, while black-colored edges show the interconnectivity within PPIN-2. Nodes D, E and F, are the detected spreader nodes of PPIN-1, whereas nodes 1, 4, 5, 19 and 24 are the detected spreader nodes of PPIN2. Green-colored spreader edges (i.e., edges connected with spreader nodes) show the interconnectivity between PPIN-1 and PPIN-2. The thickness of the edges varies with the order of ranking.

[Figure]

Figure 3. **Mechanism of transmission of viral infection.** SARS-CoV spreaders establish interaction with human spreader proteins, and the viral infection gets mediated from SARS-CoV PPIN to human PPIN through them. **A**. PPIN of SARS-CoV (red) in which spreader nodes are marked as blue. **B.** Interaction of SARS-CoV spreaders with its level-1 corresponding proteins in human PPIN (marked as green). **C.** Selection of spreaders in level-1 (level-1 spreaders are marked as yellow) and level-2 human proteins (level-2 spreaders are marked as green). Rest proteins in human PPIN are ignored to prevent overlap in the diagram.

[Figure]

Figure 4. **SARS-CoV PPIN.** The PPIN consists of the interaction between SARS-CoV proteins only. It is a collection of 9 SARS-CoV proteins only (marked as red).

[Figure]

Figure 5 **SARS-CoV-human PPIN (level-1).** The PPIN consists of the interaction between SARS-CoV and human proteins. The blue node represents SARS-CoV spreaders, while the yellow node represents SARS-CoV s level-1 human spreaders. The thickness of the edges varies with the order of ranking.

[Figure]

Figure 6 **SARS-CoV-human PPIN (level-1 and level-2).** The PPIN consists of the interaction between SARS-CoV and human proteins. The blue node represents SARS-CoV spreaders, while the yellow and green nodes represent SARS-CoV s level-1 and level-2 human spreaders. The thickness of the edges varies with the order of ranking.

*5. Line 308-309: the complete network has been generated online. Specific tool and parameters used to construct the network should be mentioned in the methods section.*

**Reply:** The entire PPIN (as mentioned in Line 308-309) is developed by coding in python by using pyvis module. Then it is hosted online directly. Pyvis module is freely available for creating interactive network graphs. The same has been highlighted in red in the manuscript.

*6. Discussion part should be elaborated more. For example, how do the most spreader nodes (at least top 3 or top 5 spreader nodes) relate to COVID19. How does the network in Figure 4, 5 and 6 relate to COVID19?*
*- Can you highlight the significance of the network, proteins and edges identified in this study with COVID19?*

**Reply:** The current work focuses only on identifying spreader nodes and edges in SARS-CoV PPIN because of the high genetic similarity of SARS-CoV and SARS-CoV-2. But with the gradual progress, the same has also been applied to form SARS-CoV2-Human PPIN in our next research work, which is available as a preprint in arXiv [1]. Here, human spreader nodes are identified by the spreadability index in the SARS-CoV-human interaction PPIN, which is also validated by Susceptible-Infected-Susceptible (SIS) disease model. This serves as an input to a unique fuzzy protein-protein interaction model along with UniProt collected n-CoV proteins. It generates level 1 human spreaders of n-CoV. Hence its level 2 human spreaders are identified using the spreadability index as implemented earlier. Once the SARS-CoV2-Human PPIN is formed, drug repurposing of COVID-19 has also been extensively studied in another work of ours which is also available as a preprint in OSF [2]. The same PPIN has also been compared [2] with the SARS-CoV2-Human PPIN generated in the work of Gordon et al. (10.1038/s41586-020-2286-9) [3]. All the above information is now added in the Conclusion section. Please see the lines marked in red.

**References:**

1. S. Saha, A.K. Halder, S.S. Bandyopadhyay, P. Chatterjee, M. Nasipuri, S. Basu, Computational modeling of Human nCoV protein-protein interaction network, ArXiv (2020). https://arxiv.org/abs/2005.04108
2. Saha, S., Halder, A. K., Bandyopadhyay, S. S., Chatterjee, P., Nasipuri, M., Bose, D., & Basu, S. (2020, May 12). Is Fostamatinib a possible drug for COVID-19? – A computational study. https://doi.org/10.31219/osf.io/7hgpj
3. Gordon DE, Jang GM, Bouhaddou M, Xu J, Obernier K, White KM, et al. A SARS-CoV-2 protein interaction map reveals targets for drug repurposing. Nature 2020; 583:459–68. https://doi.org/10.1038/s41586-020-2286-9.

*7. How does the developed network attribute, spreadability index, and the generated SIS model perform better than existing approaches? How do you know SIS model performs better than other approaches? @ as any testing carried out? Please explain.*
*a. Discussion and conclusion parts can be separated.*

**Reply:** Synthetic PPINs (Figure 1) are used first for testing our proposed methodology. Then, the same is also compared to the other existing methods like DC, BC, CC, and LAC, as shown in Table 1 to Table 5. The results show satisfactory results.

We have studied two models in models in epidemiology: SIR and SIS.

**SIR:**

**S:** The number of susceptible individuals. When a susceptible and an infectious individual come into "infectious contact", the susceptible individual contracts the disease and transitions to the infectious compartment.

**I:** The number of infectious individuals. These are individuals who have been infected and are capable of infecting susceptible individuals.

**R:** The number of removed (and immune) or deceased individuals. These individuals have been infected and have either completely recovered from the disease and entered the removed compartment (completely immune to the disease), or died. It is assumed that the number of deaths is negligible concerning the total population. This compartment may also be called "recovered" or "resistant". They will not be again infected with the same disease.

**SIS:**

**S:** The number of susceptible individuals. When a susceptible and an infectious individual come into "infectious contact", the susceptible individual contracts the disease and transitions to the infectious compartment.

**I:** The number of infectious individuals. These are individuals who have been infected and are capable of infecting susceptible individuals.

**S:** Individuals get recovered and again become susceptible.

Out of SIR (susceptible, infected, and recovered) and SIS model (susceptible, infected, and susceptible), SIS is the standard one because, from the perspective of viral infection, there is nothing called a completely recovered state (**R**) because if someone gets infected with viral disease, then they will avail drugs for the disease and will get cured. But they again become susceptible to the same disease (**S**). Therefore, it cannot be guaranteed that the viral infection will not happen again after getting cured. So, the SIS model is given more priority than SIR in this proposed work. Besides, the SIS model is used only to validate the spreader nodes identified by the spreadability index. It can be observed from Table 1 to Table 5 that the proposed methodology has the highest SIS interaction rate of 2.46 with viral proteins (see Table 1) in comparison to others for their corresponding top 10 spreader nodes in the synthetic PPIN, as shown in Figure 1.

The discussion and conclusion section is now separated and updated. Please see the changes marked in red.

*8. The background is well structured and written. However, the authors need to improve on other sections, as followed:*
*- Please improve on figures and all figure captions.*
*- There are grammatical mistakes and typo errors throughout the manuscript.*
*- Several points, such as data collection and the network construction (tools, parameters in constructing the network), need to be addressed in the methods.*
*- The organization of results need to be improved.*
*- The results need to be critically discussed and concluded.*
*- Please be consistent with the term of PPIN and network. Several terms need to be clearly defined.*

**Reply:** Figures and all figure captions are improved now. A full effort has been made to eradicate grammatical mistakes and typo errors throughout the manuscript. Data collection and PPIN construction tools have been included. The result, discussion, and conclusion sections are updated. Consistency with the term of PPIN is ensured in all the sections. Several terms have been included in the Theory & Notations section for better readability. All changes are marked in red in the manuscript.